# Sex-Related Differences in Patient Selection for and Outcomes after Pace and Ablate for Refractory Atrial Fibrillation: Insights from a Large Multicenter Cohort

**DOI:** 10.3390/jcm11164927

**Published:** 2022-08-22

**Authors:** Thomas Baumgartner, Miriam Kaelin-Friedrich, Karol Makowski, Fabian Noti, Beat Schaer, Andreas Haeberlin, Patrick Badertscher, Nikola Kozhuharov, Samuel Baldinger, Jens Seiler, Stefan Osswald, Michael Kühne, Laurent Roten, Hildegard Tanner, Christian Sticherling, Tobias Reichlin

**Affiliations:** 1Department of Cardiology, Inselspital University Hospital Bern, University of Bern, 3010 Bern, Switzerland; 2Department of Cardiology, University Hospital Basel, 4031 Basel, Switzerland; 3Department of Internal Medicine, Kantonsspital Winterthur, 8400 Winterthur, Switzerland

**Keywords:** pace and ablate, AV junction ablation, sex-related differences

## Abstract

**Background:** A pace and ablate strategy may be performed in refractory atrial fibrillation with rapid ventricular response. **Objective:** We aimed to assess sex-related differences in patient selection and clinical outcomes after pace and ablate. **Methods:** In a retrospective multicentre study, patients undergoing AV junction ablation were studied. Sex-related differences in baseline characteristics, all-cause mortality, heart failure (HF) hospitalizations, and device-related complications were assessed. **Results:** Overall, 513 patients underwent AV junction ablation (median age 75 years, 50% men). At baseline, men were younger (72 vs. 78 years, *p* < 0.001), more frequently had non-paroxysmal AF (82% vs. 72%, *p* = 0.006), had a lower LVEF (35% vs. 55%, *p* < 0.001) and more frequently had cardiac resynchronization therapy (75% vs. 25%, *p* < 0.001). Interventional complications were rare in both groups (1.2% vs. 1.6%, *p* = 0.72). Patients were followed for a median of 42 months in survivors (IQR 22–62). After 4 years of follow-up, the combined endpoint of all-cause death or HF hospitalization occurred more often in men (38% vs. 27%, *p* = 0.008). The same was observed for HF hospitalizations (22% vs. 11%, *p* = 0.021) and all-cause death (28% vs. 21%, *p* = 0.017). Sex category remained an independent predictor of death or HF hospitalization after adjustment for age, LVEF and type of stimulation. Lead-related complications, infections, and upgrade to ICD or CRT occurred in 2.1%, 0.2% and 3.5% of patients, respectively. **Conclusions:** Pace and ablate is safe with a need for subsequent device-related re-interventions in 5.8% over 4 years. We found significant sex-related differences in patient selection, and women had a more favourable clinical course after AV junction ablation.

## 1. Introduction

Atrial fibrillation has shown a steep increase in recent years and is likely to further increase [1]. Despite significant advantages in catheter ablation [2], rhythm control by means of catheter ablation and/or medication is not attainable in a substantial proportion of patients with atrial fibrillation, in particular in patients with persistent, long standing and permanent atrial fibrillation [3,4]. For those with intractable symptoms or insufficient rate control, pace and ablate is a well-established treatment option, which is technically straight-forward and frequently results in immediate symptomatic benefit [5,6,7,8].

The pace and ablate approach received significant attention, when the recent APAF-CRT study showed that cardiac resynchronization therapy (CRT) followed by AV junction ablation improved all-cause mortality as well as the combination of all-cause mortality and heart failure (HF) hospitalizations, when compared to pharmacological therapy in patients with permanent atrial fibrillation and narrow QRS [5,9]. However, uncertainties regarding optimal patient selection and timing for this procedure remain. One of the concerns is the risk for subsequent device-related complications in pacemaker-dependent patients [10,11]. However, with a sample size of 63 patients in the intervention arm, APAF-CRT was too small to study the incidence of device-related complications in this particularly vulnerable subset of patients [9].

Sex- and gender-related bias in patient selection as well as differences in outcomes have been described for numerous medical procedures [12,13,14,15,16,17]. The impact on patient selection for and outcome after pace and ablate has not been studied to date. Therefore, we aimed to assess sex- and gender-related differences in patient selection and clinical outcomes in a large multicentre cohort of patients undergoing AV junction ablation for refractory atrial arrhythmias.

## 2. Methods

### 2.1. Study Population

Consecutive patients undergoing catheter ablation in Switzerland are prospectively enrolled into a national ablation registry [18]. For this retrospective multicentre analysis, consecutive patients undergoing AV junction ablation as part of a pace-and-ablate strategy for refractory atrial fibrillation between 2011 and 2019 at the two largest ablation centres in Switzerland were enrolled. Patient data were collected from the electronic health records systems. Patients with missing follow-up data were excluded from analysis.

The study was performed in accordance with the principles of the Declaration of Helsinki. The study protocol was approved by local Ethics Committees. The authors had full access to and take full responsibility for the integrity of the data.

### 2.2. Baseline Evaluation

All patients underwent pre-procedural clinical evaluation including detailed medical history and standard blood tests. A detailed device history was obtained, including the type of device (pacemaker or ICD), the type of stimulation (RV only or biventricular), and the time since first device-implant. Transthoracic echocardiography was performed to assess the left ventricular ejection fraction (LVEF).

### 2.3. Ablation Procedure

The AV junction ablation procedures were performed in local anaesthesia and guided by fluoroscopy. The 3D electro-anatomical mapping systems were only exceptionally used. The selection of a non-irrigated or irrigated ablation catheter was based on the operator’s discretion.

### 2.4. Follow-Up

Follow-up data and vital status were obtained from the electronic health records. Hospitalisation for heart failure was adjudicated by the local investigators.

The primary endpoint of our analysis was a composite clinical outcome consisting of all-cause mortality or hospitalisation for heart failure during follow-up. Secondary endpoints were the individual components of the primary endpoint (all-cause mortality and hospitalisation for heart failure), device-related re-interventions during follow-up due to lead-complications, infections, or an upgrade to ICD or CRT.

### 2.5. Statistical Analysis

Categorical variables are reported as frequencies and percentages, for frequency comparisons we used chi-square tests. Continuous variables are reported as medians and interquartile ranges (IQR), comparisons between groups were performed with the Mann–Whitney U test. We additionally performed univariate and multivariate Cox regression models to investigate associations between sex category and outcomes. These associations are reported as hazard ratios (HRs) with 95% confidence intervals (CIs). We further illustrate incidences of endpoints using Kaplan–Meier curves, dichotomized by sex category. A *p*-value of <0.05 was considered statistically significant. Statistical analyses were performed using SPSS Statistics, IBM Corp. Released 2017. IBM SPSS Statistics for Windows, Version 25.0, Armonk, NY: IBM Corp.

## 3. Results

### 3.1. Study Population

Between 2011 and 2019, 513 AV junction ablations were performed in the two participating centres. Baseline and procedural characteristics are summarized in Table 1 and Table 2. Median age was 75 years (IQR 69–81), and 258 (50%) patients were women. Baseline differences between men and women were observed for age (72 vs. 78 years, *p* < 0.001), comorbidities such as coronary artery disease (47% vs. 24%, *p* < 0.001), and for the presence of persistent AF (82% vs. 72%, *p* < 0.001). No significant difference was present concerning symptom status with 48% of patients having NYHA class III or greater and 62% having EHRA class III or greater. LVEF was significantly lower in men at baseline (35% vs. 55%, *p* < 0.001) and median QRS-duration was significantly longer (122 ms vs. 96 ms, *p* < 0.001). While Beta-blocker and ACEI/ARB therapy were more frequently prescribed in men (88% vs. 81%, *p* = 0.03 and 71% vs. 63%, *p* = 0.04, respectively), no significant differences were noted among other medications.

### 3.2. Pace and Ablate as Primary or Secondary Treatment Strategy

Patient selection for primary ablation of the AV junction (without any preceding attempt at interventional rhythm control by means of pulmonary vein isolation) or secondary ablation of the AV junction (after failed previous attempts for atrial fibrillation ablation) was not different between men and women (primary strategy 76% vs. 77%, secondary strategy 24% vs. 23%, *p* = 0.94).

### 3.3. Device Types Used for Pace and Ablate

In 359 cases (70%), a pacemaker or ICD had been implanted in a previous hospitalization. In the remaining cases, device implantation was performed immediately before AV junction ablation. Overall, CRT devices were more frequently implanted in men compared to women (53% vs. 17%, *p* < 0.001), as were ICDs (31% vs. 6%, *p* < 0.001).

### 3.4. Outcomes after Pace and Ablate

Patients were followed for a median of 42 months in survivors (IQR 22-62). After 4 years of follow-up, the primary endpoint of all-cause death or HF hospitalization had occurred in 32% of patients (Figure 1). The primary endpoint occurred significantly more frequently in men (38% vs. 27%, *p* = 0.008). The individual components of the primary endpoint showed a similar association with sex category: all-cause death occurred in 25% of the patients and was more common in men compared to women (28% vs. 21%, *p* = 0.017); HF hospitalizations occurred in 16% of the patients and were also more common in men compared to women (22% vs. 11%, *p* = 0.021); (Figure 2).

Male sex category remained an independent predictor of the combined endpoint of death or HF hospitalization after adjustment for age, LVEF and type of stimulation (HR 1.45 [1.0–2.1], *p* = 0.047, Table 3).

### 3.5. Device-Related Re-Interventions

During follow-up, device-related re-interventions occurred in 5.8% of the patients: lead-related complications in 11 patients (2.1%) after a median of 362 days, infections in 1 patient (0.2%) after 1142 days, and upgrade to an ICD or CRT in 18 patients (3.5%) after a median of 238 days (Figure 3). No sex-related differences were observed (all *p*-values > 0.05).

## 4. Discussion

This study assessed sex- and gender-related differences in patient selection and long-term clinical outcomes in a large multicentre cohort of 513 patients undergoing AV junction ablation as part of a pace and ablate strategy for refractory atrial fibrillation. We report the following major findings:

First, the patient characteristics of men as compared to women chosen for pace and ablate differed significantly in our cohort. Men were younger, had a lower LVEF and a higher burden of comorbidities. Second, no sex and gender differences were observed in the use of pace and ablate as a primary treatment strategy or secondary after a prior failed atrial fibrillation ablation. Third, the primary endpoint of all-cause death or HF hospitalization occurred significantly more frequently in men compared to women during 4 years of follow-up (38% vs. 27%, *p* = 0.008). The better outcome in women was independent of age, LVEF and mode of stimulation. Fourth, the rate of device-related complications during 4 years of follow-up was 5.8%.

Our findings have clinical implications: In light of the ongoing increase in the worldwide incidence of atrial fibrillation, therapeutic options are necessary for patients in whom rhythm control or adequate rate and symptom control by means of medication and or catheter ablation cannot be achieved. For this patient group, the pace-and-ablate strategy is a well-established, though irreversible treatment option. The recent APAF-CRT trial showed that biventricular pacing and ablation was superior to pharmacological therapy in reducing mortality in patients with permanent AF [9]. In line with large registry data from Sweden [19], the mortality of AF patients in the control arm of APAF-CRT was over 40% after 4 years [9]. Using biventricular pacing and AV junction ablation, mortality could be reduced to 14% in APAF-CRT. The benefit of rate control and rate regularization provided by the pace and ablate strategy was confirmed in our study. The slightly higher mortality of 25% after 4 years observed in our cohort as similar to other studies [20,21] might be explained by two factors. First, the use of biventricular stimulation was only 35% in our cohort, and the non-physiological RV pacing likely has contributed to the higher mortality [22]. Second, the patients in our cohort were, on average, 3 years older compared to the intervention group in APAF-CRT [9].

The rate of device-related early adverse events reported in APAF was 4.8% and consisted of lead dislodgements [9]. However, with only 63 patients in the intervention arm, the study was too small to provide information on the incidence of device-related re-interventions during follow-up. Based on large registry data from Scandinavia, the rate of major complications was already 5.6% after 6 months [11]. Given that device-related complications in pacemaker-dependent patients after AV junction ablation have a significant morbidity and mortality, the finding of only 5.8% device-related reinterventions after 4 years in our study was lower than expected and reassuring [10,11]. Furthermore, the majority of those reinterventions (3.5%) were upgrades without immediate device-related risk for the patients. Nevertheless, optimal implantation techniques (including the use of biventricular pacing or conduction system pacing) and measures to minimize device infections are important for all patients in need of a pacemaker or an ICD, but should particularly be considered for patients planned for a pace-and-ablate strategy [20,23,24].

The body of evidence about sex- and gender-related differences, but also sex- and gender-related bias in medicine and also cardiology is growing [12,13,14,15,16,17]. The relative contributions of sex-related (biology) and gender-related (behaviour) differences to the observed overall differences are often difficult to separate. The classification of patients in our study was based on sex. However, we cannot rule out that the observed differences were also partially affected by gender as well. In a recent study, Mohamed and co-workers reported that women are more frequently selected for CRT-P and men more frequently for CRT-D [16]. In our study, the selection of pace and ablate as a primary or secondary strategy (after failed PVI) did not differ between men and women. However, men significantly more often received biventricular simulation compared to women (53% vs. 17%). Given the significant differences in baseline characteristics, women and men in our study seemed to correspond to two main categories of patients who received a pace and ablate treatment: one category of patients with relatively few comorbidities and intolerable AF symptoms that were more often found in women, and another category of patients with a high burden of comorbidities and low LVEF more commonly found in men. In addition, women seem to be affected by more adverse effects of rate control medication as betablockers, in particular substances that are metabolised via Cyp2D6 [25]. These findings might reflect a sex-related bias in patient selection as well as differences in atrial fibrillation phenotypes in patients with heart failure with reduced ejection fraction and heart failure with preserved ejection fraction. Of important note, male sex category was independently associated with a negative clinical outcome even after adjustment for age, LVEF and type of stimulation. This, to our knowledge, is a new finding and necessitates further investigations.

Conduction system pacing by means of His bundle pacing or left bundle pacing has already remarkably affected the field of ventricular pacing and will do so even more so in the upcoming years. Given the advantages of conduction system pacing such as a shorter QRS width, more favourable LV remodelling compared to CRT [26,27] will likely make this the preferred approach in patients undergoing pace and ablate in the future [28].

## 5. Limitations

Potential limitations of the present study merit consideration. First, given the significant differences in baseline characteristics, residual confounding cannot be excluded and may have had some impact. Second, given that patient identification occurred via a prospective national ablation registry [18], the risk for introduction of a relevant bias seems low. Third, despite a sample size that was more than eight times larger than that of the intervention group in APAF-CRT, the number of patients with device-related re-interventions was too low to assess for predictors. Fourth, the lead position for RV and LV pacing was based on the implanting physician’s preference and, hence, was heterogeneous within the study sample. Since leads position may clearly impact on outcomes, this might have introduced a small bias.

## 6. Conclusions

A pace and ablate strategy in patients with refractory atrial fibrillation is safe. Subsequent device-related re-interventions are needed in 5.8% over 4 years. We found significant sex and gender differences in patient selection. Even after adjustment for age, LVEF and type of stimulation, women had a more favourable clinical course after AV junction ablation.

## Figures and Tables

**Figure 1 jcm-11-04927-f001:**
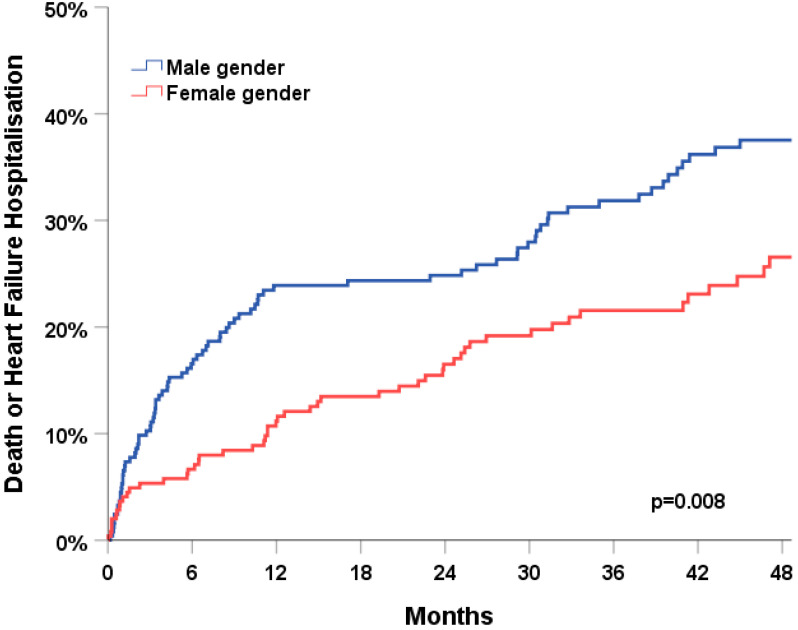
Association of sex category with the combined clinical endpoint of all-cause death or heart failure hospitalisation during 4 years of follow-up.

**Figure 2 jcm-11-04927-f002:**
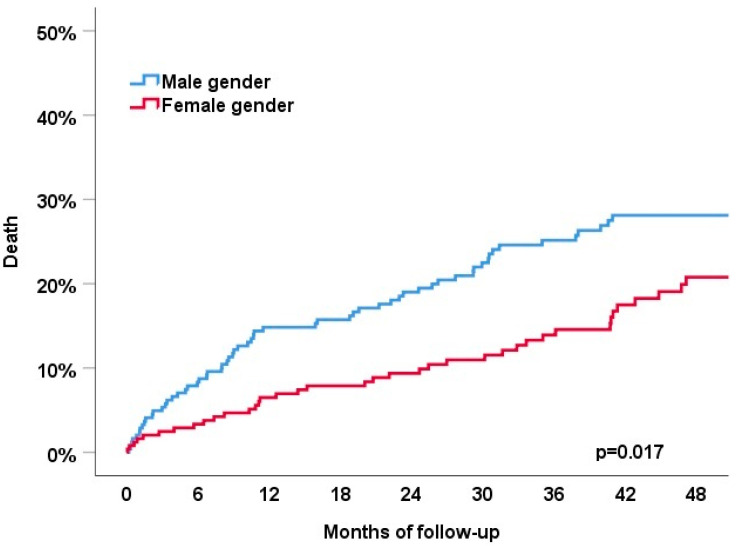
Association of sex category with the all-cause death (**top**) and heart failure hospitalization (**bottom**) during 4 years of follow-up.

**Figure 3 jcm-11-04927-f003:**
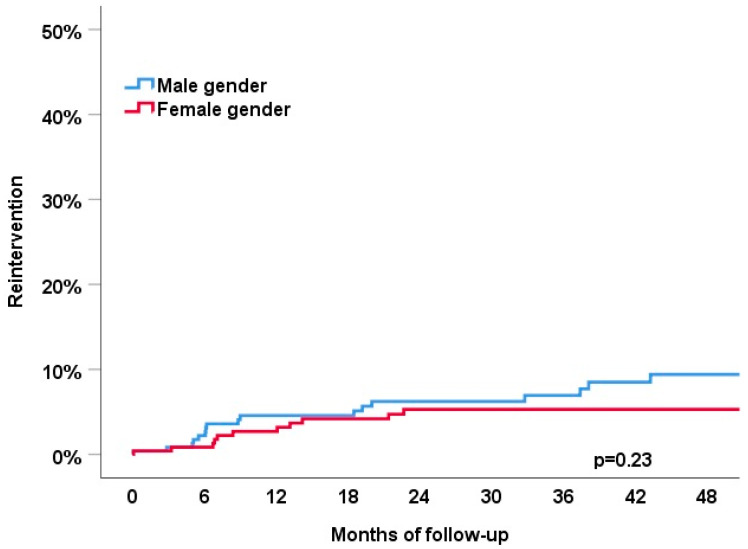
Association of sex category with the device-related reinterventions (lead failure, infections or upgrade to ICD or CRT) during 4 years of follow-up.

**Table 1 jcm-11-04927-t001:** Baseline characteristics of the patients.

	All Patients (*n* = 513)	Men (*n* = 255)	Women (*n* = 258)	*p* Value
Age, median (years)	75 (69–81)	72 (66–78)	78 (73–82)	<0.001
Duration of AF, median (months)	93 (50–151)	98 (49–158)	89 (50–142)	0.21
Paroxysmal (*n*, %)	117 (23%)	45 (18%)	72 (28%)	<0.001
Persistent (*n*, %)	396 (77%)	210 (82%)	186 (72%)	
Primary AVJ-Ablation (*n*, %)	393 (77%)	195 (76%)	198 (77%)	0.94
Secondary AVJ-Ablation (*n*, %)	120 (23%)	60 (24%)	60 (23%)	0.94
Number of previous PVI in pts with secondary AVJ ablation, median	2 (1,3)	2 (1,3)	2 (1,2)	0.11
Coronary artery Disease (*n*, %)	184 (36%)	121(47%)	63 (24%)	<0.001
Diabetes Mellitus (*n*, %)	104 (20%)	67 (26%)	37 (14%)	0.001
Hypertension (*n*, %)	368 (72%)	173 (68%)	195 (76%)	0.05
NYHA class ≥3 (%)	246 (48%)	133 (52%)	113 (44%)	0.06
EHRA class ≥3 (%)	317 (62%)	155 (61%)	162 (63%)	0.68
QRS, median (ms)	104 (90–142)	122 (96–151)	96 (84–120)	<0.001
LVEF, median (%)	46 (30–60)	35 (25–53)	55 (45–60)	<0.001
Beta-blockers (%)	433 (84%)	224 (88%)	209 (81%)	0.03
ACEI/ARB (%)	344 (67%)	182 (71%)	162 (63%)	0.04
Digoxin (%)	72 (14%)	35 (14%)	37 (14%)	0.4
Amiodarone (%)	127 (25%)	67 (26%)	60 (23%)	0.4
Other antiarrhythmic drugs (%)	24 (5%)	6 (2%)	18 (7%)	0.01
Prior device present (*n*, %)	359 (70%)	187 (74%)	172 (67%)	0.08
Any prior issue (*n*, %)	20 (4%)	14 (6%)	6 (2%)	0.18
Biventricular Stimulation (*n*, %)	179 (35%)	134 (53%)	45 (17%)	<0.001
RV-Stimulation (*n*, %)	332 (65%)	120 (47%)	212 (82%)	
ICD (*n*, %)	95 (19%)	80 (31%)	15 (6%)	<0.001
PM only (*n*, %)	419 (82%)	175 (69%)	243 (94%)	

AVJ = atrioventricular junction; PVI = pulmonary vein isolation; primary AVJ-ablation = without preceding PVI; secondary AVJ-ablation = after failed PVI (s); NYHA = New York Heart Association; EHRA = European Heart Rhythm Association; LVEF = left ventricular ejection fraction; RV = right ventricle; ICD = implantable cardioverter defibrillator; PM = pacemaker.

**Table 2 jcm-11-04927-t002:** Procedural characteristics.

	All Patients (*n* = 513)	Men (*n* = 255)	Women (*n* = 258)	*p* Value
Primary AVJ-Ablation (*n*, %)	393 (77%)	195 (77%)	198 (77%)	0.94
Secondary AVJ-Ablation (*n*, %)	120 (23%)	60 (23%)	60 (23%)	0.94
Number of previous PVI, median	2 (1,3)	2 (1,3)	2 (1,2)	0.11
Procedure time, median (minutes)	40 (30–60)	40 (30–60)	41 (30–60)	0.81
Pace and ablate in one procedure (*n*, %)	151 (30%)	66 (26%)	85 (33%)	0.08
Primarily successful AVJ-Ablation (*n*, %)	503 (98%)	248 (97%)	255 (99%)	0.2
Any Interventional complication (*n*, %)	7 (1%)	3 (1%)	4 (2%)	0.72

AVJ = atrioventricular junction.

**Table 3 jcm-11-04927-t003:** Univariate and multivariable Cox regression analysis to predict the primary endpoint of death or HF hospitalization.

	Univariate Analysis		Multivariable Analysis	
	Hazard Ratio	*p*-Value	Hazard Ratio	*p*-Value
Male sex	1.54 [1.12–2.11]	0.008	1.45 [1.00–2.08]	0.047
LVEF, per % increase	0.98 [0.97–0.99]	<0.001	0.97 [0.96–0.98]	<0.001
Age, per year	1.02 [1.00–1.04]	0.049	1.03 [1.01–1.05]	0.003
RV-only Stimulation	0.78 [0.57–1.08]	0.14	1.59 [1.04–2.42]	0.032

LVEF = left ventricular ejection fraction; RV = right ventricle.

## Data Availability

The utilized data is available from the corresponding author on request.

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
