# Peer review of "Sex-Related Differences in Patient Selection for and Outcomes after Pace and Ablate for Refractory Atrial Fibrillation: Insights from a Large Multicenter Cohort"

_jcm, 2022, doi:10.3390/jcm11164927_

Round 1
Reviewer 1 Report
Baumgartner and colleagues present data from a retrospective registry on outcomes after AV-nodal ablation. The presentation is coherent, the findings are of great interest.
However, I have following minor concerns:
- The methods section could be improved. The authors specify, that incomplete datasets were excluded, what was the exclusion rate? Was ablation successful in all included cases?
- Table 1: Definitions of primary/secondary AVN ablation should be added to the table legend, although they are defined later in the text. Two seperate tests between binary parameters are obsolete (ICD/PM).
- Discussion: Authors describe sex and gender related differences and even discuss that the observed differences were certainly affected by gender as well. However, the manuscript presents data from hospital records most probably reflecting sex (biological). If no data on gender (social identification) is available, this assumption should not be made.
Author Response
Comments Reviewer #1:
Baumgartner and colleagues present data from a retrospective registry on outcomes after AV-nodal ablation. The presentation is coherent, the findings are of great interest.
However, I have following minor concerns:
- The methods section could be improved. The authors specify, that incomplete datasets were excluded, what was the exclusion rate?
Missing follow-up information was the only exclusion criterion. This has been specified in the methods section.
Methods section, page 2, line 21
- Was ablation successful in all included cases?
As reported in Table 2, AV node ablation was acutely successful in 98%.
Table 2
- Table 1: Definitions of primary/secondary AVN ablation should be added to the table legend, although they are defined later in the text.
The definitions have been added to the table legend of Table 1.
Table 1
- Table 1: Two seperate tests between binary parameters are obsolete (ICD/PM).
This has been changed according to the suggestion of the reviewer.
Table 1
- Discussion: Authors describe sex and gender related differences and even discuss that the observed differences were certainly affected by gender as well. However, the manuscript presents data from hospital records most probably reflecting sex (biological). If no data on gender (social identification) is available, this assumption should not be made.
We agree with the reviewer. The wording was changed to “We can however not rule out that the observed differences were also partially affected by gender as well.”
Discussion, page 7, line 48f
Reviewer 2 Report
I would congratulate for this very good paper, where authors aimed to assess sex-related differences in patient selection and clinical outcomes after pace and ablate. Authors found significant sex-related differences in patient selection and how women had a more favorable clinical course after AV-junction ablation. Results are very interesting, here you have some minor comments in order to improve the manuscript:
Introduction, in the line: “Despite significant advantages in catheter ablation”: As advantages I would focus on the modern advantages of awareness of radiation doses and risks, also during 2022 interventional cardiology procedures, that is essential today in order to apply the risk-benefit assessment, a prerequisite to create a culture of respect for radiation hazard and a commitment to minimise exposure and maximise protection ( please cite: DOI: 10.1080/00015385.2020.1733303)
Despite this, “ rhythm control by means of catheter ablation and/or medication is not attainable in a substantial proportion of patients with atrial fibrillation”: I would add specially in persistent, long standing and permanent AF forms
In Methods and Results, among RV or biventricular stimulation, authors should clarify about leads positions for RV (septum or apex?) and LV (which percentage of “target” vein? Anterior? Anterolateral? Lateral? Posterior?) Please discuss more about this, since leads position may clearly impact on outcomes in patients undergoing AV junction ablation
Sleep-disordered breathing tipically occurs in HF patients. Did authors consider this parameter in their patient population? Not surprisingly, a correlation between CRT response and sleep apnoea burden considering gender differences has been documented. In particular, HF-women responders to CRT demonstrate a significant linear decrease in sleep apnoea burden determined through a device algorithm, when compared to a similar male population (please cite DOI: 10.1016/j.sleep.2019.06.019)
In discussion authors should also cite in 2022 the existence of His bundle pacing (HBP), alone or optimized in association with coronary sinus pacing (HBP+LV), that has recently been proposed as an alternative to conventional cardiac resynchronization therapy (CRT) (doi: 10.1016/j.jacc.2018.09.073) . In particular, compared with conventional CRT, a shorter QRS width can be obtained with HBP alone or in association with coronary sinus pacing but today we are unable to show a better clinical outcome and there is urgent need for large, randomized trials, also considering sex related differences (DOI: 10.1111/pace.14336). Please discuss and cite both suggested references
Author Response
Comments Reviewer #2:
I would congratulate for this very good paper, where authors aimed to assess sex-related differences in patient selection and clinical outcomes after pace and ablate. Authors found significant sex-related differences in patient selection and how women had a more favorable clinical course after AV-junction ablation. Results are very interesting, here you have some minor comments in order to improve the manuscript:
- Introduction, in the line: “Despite significant advantages in catheter ablation”: As advantages I would focus on the modern advantages of awareness of radiation doses and risks, also during 2022 interventional cardiology procedures, that is essential today in order to apply the risk-benefit assessment, a prerequisite to create a culture of respect for radiation hazard and a commitment to minimise exposure and maximise protection ( please cite: DOI: 10.1080/00015385.2020.1733303)
The reference has been added as suggested by the reviewer.
Introduction, page 1, line 39
- Despite this, “ rhythm control by means of catheter ablation and/or medication is not attainable in a substantial proportion of patients with atrial fibrillation”: I would add specially in persistent, long standing and permanent AF forms
We fully agree with the reviewer and this has been added accordingly.
Introduction, page 1, line 41
- In Methods and Results, among RV or biventricular stimulation, authors should clarify about leads positions for RV (septum or apex?) and LV (which percentage of “target” vein? Anterior? Anterolateral? Lateral? Posterior?) Please discuss more about this, since leads position may clearly impact on outcomes in patients undergoing AV junction ablation
The lead position was based on the implanting physician’s preference and hence was heterogeneous within the study sample. We however do not have the data on the exact lead positions and can not provide that data.
We added this as a limitation to the discussion section.
Discussion, Limitations, page 8, line 19ff
- Sleep-disordered breathing tipically occurs in HF patients. Did authors consider this parameter in their patient population? Not surprisingly, a correlation between CRT response and sleep apnoea burden considering gender differences has been documented. In particular, HF-women responders to CRT demonstrate a significant linear decrease in sleep apnoea burden determined through a device algorithm, when compared to a similar male population (please cite DOI: 10.1016/j.sleep.2019.06.019)
We did not collect data on sleep-disordered breathing.
- In discussion authors should also cite in 2022 the existence of His bundle pacing (HBP), alone or optimized in association with coronary sinus pacing (HBP+LV), that has recently been proposed as an alternative to conventional cardiac resynchronization therapy (CRT) (doi: 10.1016/j.jacc.2018.09.073) . In particular, compared with conventional CRT, a shorter QRS width can be obtained with HBP alone or in association with coronary sinus pacing but today we are unable to show a better clinical outcome and there is urgent need for large, randomized trials, also considering sex related differences (DOI: 10.1111/pace.14336). Please discuss and cite both suggested references
We fully agree with the reviewer on the pivotal role of conduction system pacing for
ventricular pacing & cardiac resynchronization in general and also more and more in patients undergoing pace and ablate. As suggested by the reviewer, we have added a paragraph on conduction system pacing to the discussion section.
Discussion, page 8, line 12ff
Round 2
Reviewer 2 Report
Manuscript definitely improved after reviewer's suggestions. Congratulations to authors